

# Heterogeneous OH oxidation of secondary brown carbon aerosol

Elijah G. Schnitzler, Jonathan P. D. Abbatt

Department of Chemistry, University of Toronto, Toronto, ON, M5S 3H6, Canada

*Correspondence to*: Jonathan P. D. Abbatt (jabbatt@chem.utoronto.ca)

**Abstract.** Light-absorbing organic aerosol, or brown carbon (BrC), has significant but poorly-constrained effects on climate; for example, oxidation in the atmosphere may alter its optical properties, leading to absorption enhancement or bleaching. Here, we investigate for the first time the effects of heterogeneous OH oxidation on the optical properties of a laboratory surrogate of secondary BrC in a series of photo-oxidation chamber experiments. The BrC surrogate was generated from aqueous resorcinol, or 1,3-dihydroxybenzene, and $H_2O_2$ exposed to >300 nm radiation, atomized, passed through trace gas

denuders, and injected into the chamber, which was conditioned to either 15 or 60% relative humidity (RH). Aerosol absorption and scattering coefficients and single scattering albedo (SSA) at 405 nm were measured using a photo-acoustic spectrometer. At 60% RH, upon OH exposure, absorption first increased, and the SSA decreased sharply. Subsequently, absorption decreased faster than scattering, and SSA increased gradually. Comparisons to the modelled trend in SSA, based on Mie theory calculations, confirm that the observed trend is due to chemical evolution, rather than slight changes in particle size. The initial

absorption enhancement is likely due to molecular functionalization and/or oligomerization, and the bleaching to fragmentation. By contrast, at 15% RH, slow absorption enhancement was observed, without appreciable bleaching. A multi-layer kinetics model, consisting of two surface reactions in series, was constructed to provide further insights regarding the RH-dependence of the optical evolution. Candidate parameters suggest that the oxidation is efficient, with uptake coefficients on the order of unity, and the aerosol is very viscous, even at 60% RH. At 15% RH, the aerosol will be viscous enough to

confine products of fragmentation, leading to their recombination, such that little bleaching is observed on the experimental timescale. These results further the current understanding of the complex processing of BrC that may occur in the atmosphere.

## 1 Introduction

Among all atmospheric constituents, aerosols have the most uncertain radiative forcing, partly due to an incomplete understanding of carbonaceous aerosols (Chung et al., 2012). In particular, the climate effects of light-absorbing organic

aerosol, or brown carbon (BrC) (Bond, 2001; Kirchstetter et al., 2004), are poorly constrained, compared to those of elemental black carbon (BC) (Ramanathan and Carmichael, 2008).

       One source of this uncertainty is the wide range of sources of BrC (Laskin et al., 2015). Low-temperature biomass burning results in the formation of primary BrC (Bahadur et al., 2012; Chen and Bond, 2010; Lewis et al., 2008; Radney et al., 2017). Many classes of compounds, including nitro-aromatics, polyphenols, and substituted polycyclic aromatic





hydrocarbons (Lin et al., 2016), have been identified in primary BrC, but their concentrations and absorptivities vary significantly. At a site strongly influenced by biomass burning in Germany, the contribution of nitro-aromatics to the absorption of 370 nm light by BrC was roughly 1 % (Teich et al., 2017); in contrast, in a field campaign in Israel, during a bonfire festival, the contribution of nitro-aromatics to the total absorption of >400 nm light was greater than 50 % (Lin et al.,

2017), in good agreement with results from controlled burns during the fourth Fire Lab at Missoula Experiment (Lin et al., 2016). In recent applications of size-exclusion chromatography, BrC constituents with molecular masses across the range of $10^2$–$10^4$ Da have been observed (Di Lorenzo et al., 2017; Di Lorenzo and Young, 2016; Wong et al., 2017). Larger, oligomeric compounds (> 1000 Da) have been observed to contribute most of the absorbance (Di Lorenzo and Young, 2016). If the primary BrC particles subsequently pass through a high-temperature region of biomass burning (Tóth et al., 2014), they may

form tar balls (Chakrabarty et al., 2010; Hoffer et al., 2016), which are significantly more absorptive (Alexander et al., 2008).

      BrC may also form from secondary processes. For example, ozonolysis of catechol and guaiacol, abundant emissions from biomass burning (Schauer et al., 2001), in the gas phase leads to the formation of lower-volatility products that partition into the condensed phase to form secondary organic aerosol (SOA) that is light-absorbing (Ofner et al., 2011). Light-absorbing SOA has also been observed to form from the gas-phase photo-oxidation of other precursors: e.g., naphthalene (Lambe et al.,

2013; Lee et al., 2014). In a recent field campaign, reactions in the condensed phase of cloud droplets or aqueous aerosols have also been shown to result in the formation of light-absorbing SOA (Gilardoni et al., 2016). In the laboratory, heterogeneous oxidation of catechol has been studied at air–solid (Pillar et al., 2015) and air–water (Pillar et al., 2014; Pillar and Guzman, 2017) interfaces; at the air–water interface, functionalization, leading to polyphenols and hydroxylated quinones that are expected to be highly-absorptive, followed by fragmentation was observed (Pillar and Guzman, 2017). Heterogeneous

reactions are sensitive to particle diameter, so they may in part be responsible for the observation of higher concentrations of BrC in particles in the accumulation mode than in the coarse mode (Liu et al., 2013). Reactions of nitrogen-containing species (e.g., ammonium sulfate and methylamine) with aldehydes (e.g., glyoxal and methyglyoxal) have also been shown to result in BrC (De Haan et al., 2009, 2011; Lee et al., 2013; Yu et al., 2011). Furthermore, the formation of intra- or inter-molecular charge transfer complexes, similar to what has been proposed to occur in natural waters (Del Vecchio and Blough, 2004), may

enhance the absorption of BrC (Phillips and Smith, 2014, 2015; Rincón et al., 2009).

      In addition to the wide range of classes of BrC, the evolution of BrC upon atmospheric aging contributes to the uncertainty in its climate forcing. During the lifetime of the particles or cloud droplets, BrC constituents are photolyzed or react with oxidants; the resulting chemical evolution leads to evolution of the optical properties of the aerosol. Field measurements have demonstrated that the absorption by BrC may decay drastically during transport, although a small fraction

may be recalcitrant (Forrister et al., 2015). On the other hand, in a recent field campaign, absorption at 365 nm by BrC was the same in a fresh convective storm outflow and its one-day-aged plume, suggesting either that photo-bleaching was minimal or that secondary chemistry produced new chromophores that compensated for any photo-bleaching (Zhang et al., 2017).

      In the laboratory, the evolution of absorption induced by photolysis and oxidation has been observed for a variety of BrC surrogates in the solution phase, including extracts of biomass burning BrC (Lin et al., 2016; Wong et al., 2017), extracts



of SOA derived from naphthalene under high-NO$_x$ conditions (Lee et al., 2014), products of (methyl)glyoxal and ammonium sulfate (Wong et al., 2017; Zhao et al., 2015), products of pyruvic acid polymerization (Rincón et al., 2009), and nitrophenols (Zhao et al., 2015). Fewer studies have investigated the evolution of BrC aerosol upon photolysis and heterogeneous oxidation. Aging of particle and gas emissions from biomass burning results in formation of BrC or non-absorbing SOA coatings on BC

particles, leading to enhanced absorption (Saleh et al., 2013; Tasoglou et al., 2017). In flow-tube experiments, BrC aerosol from atomized methylglyoxal and ammonium sulfate has been shown to increase in absorptivity upon exposure to ozone, due to carbonyl products (Sareen et al., 2013). BrC from biomass burning exposed to natural sunlight has been shown to photo-bleach; additionally, the rate of photo-bleaching decreased in the presence of NO$_x$, possibly due to formation of secondary BrC constituents that slightly compensate for photo-bleaching (Zhong and Jang, 2014). Most recently, in potential aerosol mass

reactor experiments, primary BrC from biomass burning exposed to irradiation, ozone, and hydroxyl radical (OH) has been shown to lose almost 50% of its absorption at 375 and 405 nm after the equivalence of 4.5 days of residence time in the atmosphere (Sumlin et al., 2017).

Here, we consider the fate of secondary BrC constituents from cloud processing upon droplet evaporation and subsequent exposure to OH radicals. The first question to raise is whether the optical properties of secondary BrC aerosol

evolve at atmospherically-relevant OH exposures; another broad question is whether this evolution is dependent on relative humidity (RH). To address these questions, we investigate for the first time the heterogeneous OH oxidation of a secondary BrC surrogate aerosol, generated from the product mixture of aqueous OH oxidation of resorcinol, or 1,3-dihydroxybenzene. Experiments were conducted in a 1-m$^3$ photo-oxidation chamber at 15 and 60% RH, and aerosol absorption, scattering, and single scattering albedo (SSA) at 405 nm were measured *in situ* using a photo-acoustic spectrometer with an integrated

reciprocal nephelometer. Changes in the optical properties were observed, raising another question, whether these changes were due to the chemical evolution of the particles or simply slight changes in the size distributions. Mie theory calculations were used to demonstrate that these changes were due to the evolution of the chemical composition and, in turn, the complex refractive index of the particles. A final question is whether the features observed for the evolution of optical properties at each RH can be reproduced by a simple kinetics model. Consequently, a multi-layer kinetics model was constructed, and potential

parameters are proposed.

## 2 Methods

### 2.1 Experimental methods

#### 2.1.1 Preparation of BrC surrogate by aqueous OH oxidation

The BrC surrogate was prepared by aqueous OH oxidation of resorcinol. An aqueous solution of 10 mM resorcinol and 10

mM H$_2$O$_2$ was prepared and placed in a cylindrical photochemical reactor (Rayonet, RPR-200), where it was irradiated with UV-B blacklights (Ushio, G8T5E) with peak emission at 306 nm. The reactor is equipped with a cooling fan and magnetic



stirrer. The solution was housed in a glass bottle, so it was not exposed to light with wavelengths <300 nm. The solution was irradiated for 4 h, resulting in an orange product mixture. When $H_2O_2$ was not added to solution, no colouration was observed in the same period of irradiation. UV-vis spectra were measured by passing light from a broadband source, with coupled deuterium and tungsten halogen lamps (Ocean Optics, DT-Mini-2), through a 1-cm quartz cuvette to a grating-based UV-vis

spectrometer (Ocean Optics, USB2000+ UV-VIS-ES). Each solution was prepared with deionized water (18 mΩ cm), and the pH was not adjusted. After irradiation, the solution was atomized immediately to inject BrC aerosol into the chamber. A fresh solution was prepared for each chamber experiment.

### 2.1.2 Preparation of yellow dye aerosol

Heterogeneous OH oxidation of a yellow azo dye, Cibacron Brilliant Yellow 3G-P (CBY; Sigma), was also investigated. The

dye has a high molecular mass, 831.02 g mol$^{-1}$, so it is non-volatile, and it contains sulfonate groups, so it is hygroscopic, similar to Solvent Black 5, which is commonly used in calibrations of aerosol optical instruments (Bluvshtein et al., 2017; Lack et al., 2006; Wiegand et al., 2014). CBY aerosol was prepared by atomizing 1 mM CBY solutions. The absorption spectrum of a diluted solution is shown in Fig. S1. Peak absorbance occurs at 404 nm.

### 2.1.3 Heterogeneous OH oxidation of BrC surrogate aerosol

The experimental setup during heterogeneous OH oxidation is illustrated in Fig. 1. The aqueous solutions of BrC surrogate were aerosolized using a constant output atomizer (TSI, 3076). To obtain appreciable aerosol absorption, it was necessary to atomize for roughly 3 h, and during this time the resorcinol product solution became significantly more absorbing. Since particles injected at the end of this period may be more absorbing than those injected at the beginning, the absorption and scattering coefficients must be qualified as time-integrated properties. Two denuders consisting of tubular mesh packed in

granular activated carbon were placed in series downstream of the atomizer to remove trace gases. In preliminary experiments without the denuders in place, significant particle growth occurred upon exposure to OH, due to gas phase photo-oxidation of volatile species in the resorcinol product mixture, increasing the scattering of the aerosol. (In the yellow dye experiments, the activated carbon denuders were not used, because the large dye did not partition out of the aerosol phase following atomization, and no particle growth occurred.)

The aerosol was then injected into the Mobile Oxidative Chamber for Aging (MOCA), which has been described in the past (Wong et al., 2015). Briefly, the chamber consists of a 1-m$^3$ bag composed of fluorinated ethylene propylene (FEP) film surrounded by an array of 24 UV-B blacklights (Philips, TL 40W/12 RS). Experiments were conducted at roughly 15 or 60% RH. Before each experiment, the chamber was cleaned by bubbling air through $H_2O_2$ (Sigma, 30% w/w in water) at about 12 L min$^{-1}$ and irradiating the chamber for at least 12 h. About 6 h before the start of each experiment, the chamber was flushed

with either dry or humidified air at 12 L min$^{-1}$, to achieve 15 or 60% RH, respectively. Temperature and RH were measured in the chamber using a capacitance probe (Vaisala, HMP75B). Air was supplied by a clean air generator (Aadco, 737).





During and after particle injection, aerosol size distributions were monitored using a scanning mobility particle sizer (SMPS), consisting of a differential mobility analyzer (DMA; TSI, 3081) and a condensational particle counter (CPC; TSI, 3772). The sample flow rate of the CPC is 1 L min$^{-1}$, but only 0.3 L min$^{-1}$ was sampled through the DMA; the difference was sampled through a filter. The DMA sheath flow rate was set to 3 L min$^{-1}$.

Aerosol absorption and scattering were measured using a photo-acoustic spectrometer (DMT, PASS), equipped with 405 and 781 nm lasers and a reciprocal integrating nephelometer (Sharma et al., 2013) with a low truncation angle (Abu-Rahmah et al., 2006). Absorption and scattering coefficients were taken as averages of 120 samples, each with an integration time of 2 s. Acoustic and zero calibrations, the latter consisting of 30 samples, were performed before each average, so the interval between averaged coefficients was 5 min. The zero calibrations account for slight changes in the gas phase background
– in particular, water vapour. The higher RH was selected to be lower than the maximum operating RH of the PASS (70%). We assume that there is no evaporation of water or organic components of the particles due to absorption (Baker, 1976; Raspet et al., 2003). In the following discussion, we characterize the optical properties of the aerosol in terms of the directly-measured absorption and scattering coefficients ($\beta_{abs}$ and $\beta_{sca}$) and the single scattering albedo ($SSA = \beta_{sca}/[\beta_{abs} + \beta_{sca}]$). Relative SSA is calculated by taking the ratio of the current and initial values; e.g., $SSA/SSA_0$.

Following aerosol injection, the size distribution and optical properties of the BrC aerosol were monitored for about 1 h before passing air through a glass bubbler containing $H_2O_2$ into the chamber at a flow rate of 1.5 L min$^{-1}$. Following 1 h of bubbling, the blacklights were turned on to produce OH from the photolysis of $H_2O_2$. Oxidation was monitored for about 3 h.

### 2.1.4 Determination of OH concentration

In a separate set of high-RH experiments, *o*-xylene (Sigma, ≥98%) was injected into the chamber as a tracer for OH, following
the aerosol injection. The concentration of the tracer was measured using a proton-transfer-reaction mass spectrometer (PTR-MS; Ionicon). Following injection of *o*-xylene, air was bubbled through $H_2O_2$ into the chamber at 1.5 L min$^{-1}$ for 1 h; subsequently, the blacklights were turned on to initiate photo-oxidation. The concentration decayed exponentially, due to dilution and pseudo-first-order reaction with OH. Dividing the experimental reaction rate constant by the second-order reaction rate constant (Atkinson and Arey, 2003) gave a steady-state OH concentration of $(1.6 \pm 0.2) \times 10^7$ molecule cm$^{-3}$. Three
experiments were performed with different initial *o*-xylene concentrations to verify that the tracer did not contribute significantly to the total OH sink; i.e., the steady-state OH concentration did not systematically increase as the *o*-xylene concentration decreased. The tracer could not be added to all experiments, because the photo-oxidation products partitioned onto the pre-existing particles to form secondary organic aerosol (SOA), which drastically increased the scattering coefficient. The average OH concentration is used to present results in terms of OH exposure, in addition to reaction time.



### 2.2 Computational methods

### 2.2.1 Mie theory calculations

To verify that changes in SSA during heterogeneous OH oxidation were induced by chemical changes to the aerosol, rather than small changes in the size distributions, we compared the observed trends to modelled trends, based on the observed size

distributions and an assumed, constant complex refractive index, $m = n + ik$, where $n$ and $k$ are the real and imaginary parts, respectively. Lognormal curves were fit to the raw geometric-cross-section-weighted size distributions (see Fig. 2a), and the absorption and scattering efficiencies for 405 nm light of individual channels between about 20 and 600 nm were calculated from the complex refractive index of the initial BrC particles, which are assumed to be spherical and homogeneously mixed, using the program Mie3Layer (Charamisinau et al., 2005). The absorption and scattering coefficients were calculated for each

channel, and total values across the distribution were used to calculate the SSA of the whole aerosol population.

### 2.2.2 Multi-layer kinetics modelling

To better understand the effects of RH on the evolution of the optical properties of the BrC surrogate, we constructed a multi-layer kinetics model based on the Pöschl-Rudich-Ammann (PRA) framework (Ammann and Pöschl, 2007; Pöschl et al., 2007). The model particles consist of a surface layer, two near-surface bulk layers, and the remaining bulk phase, as illustrated in Fig.

3. The near-surface bulk layers are included to account for concentration gradients in the bulk phase (Shiraiwa et al., 2010); each is 2 nm thick. The diameter and density of the particles are assumed to be 180 nm and 1.3 g cm$^{-3}$, respectively. The particles are assumed to be initially composed of a single BrC species, A. At the surface, A reacts with OH to form product B, and B reacts to form product C. The loss of species $i = $ A, B due to reaction with OH, $L_i$, is calculated as $\gamma_{OH,i} J_{OH} \theta_i$, where $\gamma_{OH,i}$ is the probability that a collision between OH and species $i$ leads to reaction, $J_{OH}$ is the flux of OH to the particles per

unit of surface area, and $\theta_i$ is the fractional surface coverage of species $i$ (Shiraiwa et al., 2009). To limit the degrees of freedom, we assume that $\gamma_{OH,i}$ is the same for A and B.

The rate constant for diffusion between bulk layers, $k_{b,b,i}$, is calculated as $4D_{b,i}/(\pi\delta)$, where $D_{b,i}$ is the bulk diffusion coefficient of species $i$, and $\delta$ is the layer thickness (Shiraiwa et al., 2010). For the transfer from the near-surface bulk layer 1 to the surface, $k_{b1,s,i}$, $\delta$ above is substituted with $(\delta + \delta_A)/2$, where $\delta_A$ is the effective molecular diameter of A (Shiraiwa et

al., 2010). Though B may form from oligomerization of compounds grouped into species A, we assume that the effective molecular diameters of all species are the same. We also assume the species have the same bulk diffusion coefficients, ensuring that the total concentration in the surface layer is the same as the number of surface sites. The rate constant for diffusion from the surface to the near-surface bulk layer 1, $k_{s,b1,i}$, is taken as $k_{b1,s,i}/\delta_A$.

The above considerations allow us to calculate concentrations of A, B, and C in the surface and bulk layers. However,

to compare the modelled and experimental results, we must calculate the relative absorption of the model particles. We derive the relative absorption of the particles solely from the concentrations and molar absorptivities, as in a bulk solution; i.e., the absorption that would be measured upon particle-into-liquid sampling. During heterogeneous OH oxidation, the evolution of





the SSA of the particles incorporates changes in both absorption and scattering. However, the changes in absorption are likely dominant. For example, for fixed values of $n$ (1.35) and particle diameter (200 nm), a 50% decrease in $k$ (from 0.04 to 0.02) gives a 10% increase in $Q_{sca}$ but almost a 50% decrease in $Q_{abs}$. Consequently, we assume that $Q_{sca}$ is steady, and we compare the modelled trends in relative absorption to the experimental trends in the inverse of relative SSA. We assume that the effects

of photolysis and heterogeneous OH oxidation are additive, such that the overall trend in the inverse of relative SSA can be corrected to account for photolysis.

## 3 Results and discussion

### 3.1 Initial BrC surrogate

To produce a laboratory surrogate of secondary BrC, we generated a mixture of light-absorbing products by the aqueous photo-

oxidation of resorcinol and $H_2O_2$ exposed to UV-B radiation. Initially, the solution of resorcinol and $H_2O_2$ exhibited no absorption of light at visible wavelengths, as shown in Fig. S2. Following 4 h of photo-oxidation, the solution was highly coloured, exhibiting broad absorption features that were strongly dependent on wavelength, such that absorbance across a 1-cm pathlength was about 0.6 at 400 nm but negligible at >600 nm (see Fig. S2). In the past, light-absorbing products of the aqueous photo-oxidation of resorcinol were also observed by Chang and Thompson (2010), who used 1 mM $H_2O_2$. Here, the

concentration of $H_2O_2$ was an order of magnitude greater, and the colour developed slightly faster, as illustrated by the time series of absorbance at 450 nm (see Fig. S3).

      More generally, light-absorbing products of aqueous photo-oxidation have been observed for a wide range of phenolic species, including those with methyl, methoxy, and carbonyl substituents (Chang and Thompson, 2010; Gelencsér et al., 2003; Smith et al., 2016). In detailed mechanistic studies, products of both hydroxylation and oligomerization have been identified

(Hoffer et al., 2004; Li et al., 2014; Sun et al., 2010; Yu et al., 2014, 2016). Oligomers form by C–C or C–O radical coupling (Kobayashi and Higashimura, 2003). C–C coupling might be expected to lead to greater absorption enhancement, since the resulting biphenyls and larger oligomers may have some degree of delocalization across the rings (Zhang et al., 2010). Indeed, based on density functional theory calculations, Magalhães et al. (2017) have shown that the absorptivity of bi- and terphenyls is greater in water than in the gas phase, because the planarity and delocalization increase. Interestingly, five of the ten most

abundant products of the aqueous photo-oxidation of 0.1 mM syringol with 0.1 mM $H_2O_2$ have been shown to be biphenyls (Yu et al., 2014). The concentration of the phenolic precursor used here was two orders of magnitude greater, a condition that may favour oligomerization over hydroxylation. Fragmentation may also occur during aqueous photo-oxidation, but the volatile products are removed in the activated carbon denuders, following atomization. Based on these considerations, the secondary BrC aerosol is likely composed primarily of oligomers, and we assume the average molecular mass is 326 g mol[-1],

representative of a terphenyl product of resorcinol.

      Before investigating the effects of photolysis and heterogeneous OH oxidation on the optical properties of the BrC surrogate, we performed control experiments, without either $H_2O_2$ or irradiation, to constrain the size-dependence of the optical





properties. Results from a deposition experiment at 60% RH, in which BrC was injected into the chamber and diluted with clean humidified air, are shown in Fig. 4. The initial geometric mean number diameter and standard deviation were 125 nm and 1.5, respectively, and the initial SSA was about 0.76. When the particles were unperturbed by radiation or OH, they were simply lost by dilution and deposition. Because smaller particles deposit faster than larger ones, the geometric mean surface

diameter gradually increased from about 170 to 190 nm over the course of 3 h. Both the absorption and scattering coefficients steadily decreased as particles were continuously lost, but the SSA increased slightly (see Fig. 2b) due to the increase in geometric mean surface diameter.

This observed size dependence of SSA can be compared to Mie theory calculations, based on the measured size distributions and an assumed complex refractive index. The complex refractive index should not change without photolysis or

heterogeneous OH oxidation, so we manually scanned values of $n$ and $k$ to apply for the duration of the experiment. We found that $n$ and $k$ values of 1.35 and 0.046, respectively, are suitable to reproduce the evolution of SSA in this particular experiment (see Fig. 4b). For SOA generated from anthropogenic and biogenic volatile organic compounds, $n$ values ranging from 1.35 to 1.61 have been observed (Kim and Paulson, 2013). The value of $k$ is in the regime of previous observations for BrC at 405 nm, which vary from very low values of 0.004 (Cappa et al., 2012) and 0.007 (Lack et al., 2012) to 0.112 (at 400 nm)

(Kirchstetter et al., 2004). In a recent field study in India, a similar value of 0.037 for ambient BrC was observed (Shamjad et al., 2016).

### 3.2 Evolution of BrC due to photolysis

Having constrained the size-dependence of the optical properties, based on the closure between measured and calculated SSA values, we now consider the effects of photolysis on the optical properties of the BrC surrogate. The emission of the UV-B

black-lights in the chamber is not representative of natural sunlight, so we do not draw direct comparisons between the timescale of our experiments and that of photolysis in the atmosphere. Rather, we performed photolysis experiments to account for the effects of photolysis during the heterogeneous OH oxidation experiments.

Results from a photolysis experiment at 60% RH are illustrated in Fig. 5. In each experiment, the value of $n$ was taken as 1.35, as described above; the assumption that $n$ is constant for the duration of the experiment is supported by the previous

observation that, for BrC generated from ammonium sulfate and methylglyoxal, $n$ did not change as the reaction proceeded (Tang et al., 2016). Despite closely reproducing the preparation of the BrC surrogate solution and injecting to approximately the same total particle volume in each experiment, we observed small differences in initial size distributions and SSA values, so it was necessary to vary $k$ slightly from experiment to experiment. For the calculated SSA to follow the trend observed before the lights are turned on, at relative time zero, a $k$ value of 0.040 must be adopted. When strict comparisons are made

between experiments, the relative SSA (the current SSA divided by the initial SSA) is presented. In 3 h, the geometric mean surface diameter increased from about 160 to 180 nm, similar to the deposition experiment, so little or no volatilization of the particles can be inferred. Because the diameter increased uniformly, the predicted SSA increases steadily. In contrast, the observed SSA gradually decreased once the lights were turned on, indicating that the particles slowly became more absorbing.





Photolysis has been observed to lead to absorption enhancement of other BrC surrogates in the past (Saleh et al., 2013; Wong et al., 2017; Zhao et al., 2015; Zhong and Jang, 2014). For example, the absorbance at 400 nm of water-soluble species produced by combustion of kaoliang stalk has been shown to increase during the first 30 min of irradiation in a solar simulator (Zhao et al., 2015). Recently, the peak absorption enhancement due to photolysis of biomass burning BrC constituents has been shown to increase with their molecular mass (Wong et al., 2017). Since the trace gas denuders remove volatile species after atomization, the composition of the particles is likely skewed towards an average molecular mass that is larger than that of the species in solution, possibly making the particles more susceptible to photolysis. In general, absorption enhancement may occur due to direct photolysis of chromophores, followed by radical recombination leading to larger, more absorptive oligomers, or reactions with OH from the dissociation of $H_2O_2$ generated from the photolysis of carbonyl compounds in the condensed phase (Anastasio et al., 1997; von Sonntag and Schuchmann, 1991). In the past, the absorption enhancement of biomass burning BrC exposed to natural sunlight has been shown to increase with RH (Zhong and Jang, 2014), a trend that was attributed to the increasing production and dissociation of $H_2O_2$ with increasing RH. Here, we observe similar trends in relative SSA under both low and high RH conditions (see Fig. 6b). It is unlikely that sufficient OH could be produced in the particle phase at the lower RH, so indirect photolysis of carbonyl compounds likely does not have a significant role in the observed absorption enhancement.

In past studies, the period of absorption enhancement due to photolysis has been shown to be followed by bleaching (Wong et al., 2017; Zhao et al., 2015; Zhong and Jang, 2014). In the present experiments, the irradiation time is limited by the particle losses; after about 3 h, the scattering and absorption coefficients approach the detection limits of the PASS. Near the end of the photolysis experiments, SSA levelled off, and this period may coincide with the inflection point between absorption enhancement and bleaching.

### 3.3 Evolution of BrC due to heterogeneous OH oxidation

The evolution of the optical properties of the BrC surrogate during a photo-oxidation experiment is affected by deposition, photolysis, and heterogeneous OH oxidation. Results of a representative photo-oxidation experiment at 60% RH are shown in Fig. 7; replicate experiments were performed at each RH. The initial geometric mean surface diameter was slightly higher than in the photolysis experiment described above (about 196 nm compared to 160 nm), so although the initial aerosol was slightly more scattering (higher SSA), the value of $k$ required to reproduce the SSA before irradiation is slightly higher (0.041 compared to 0.040). During irradiation, the geometric mean surface diameter first increased slightly due to deposition and then began to decrease due to volatilization. As a result, the predicted SSA first increases slightly and then, at about 90 min, begins to decrease. The observed SSA exhibited a sharp decrease within only 10 min of reaction time and an induction period before a significant increase. Both the absorption enhancement and bleaching contrast with the predicted trend of SSA, so they are certainly due to the evolution of particle composition, rather than size. The absorption enhancement was rapid enough to lead to a peak in the absorption coefficient, despite continuous particle losses (see Fig. S4).





To ensure that the sequential absorption enhancement and bleaching described above are distinctive features of the BrC surrogate, we also investigated the heterogeneous OH oxidation of a yellow dye aerosol at 60% RH. In this case, we observed uniform bleaching, as shown in Fig. S5, consistent with bulk aqueous studies of similar azo dyes (Georgiou et al., 2002). There was no decrease in the geometric mean surface diameter, so there was little or no volatilization. It is likely that

OH attacks the azo nitrogen-nitrogen bond (Hisaindee et al., 2013). The product fragments would be large enough to remain in the particle phase; for example, the smaller fragment would still contain one sulfonate group and two aromatic rings.

Unlike photolysis, heterogeneous OH oxidation is strongly dependent on RH, as shown in Fig. 6a. In contrast to the rapid absorption enhancement and bleaching observed at 60% RH, the period of absorption enhancement is prolonged at 15% RH, on a timescale similar to that of the photolysis experiments. The peak absorption enhancement at 15% RH results in a

lower relative SSA than at 60% RH. To better understand the effects of RH on heterogeneous OH oxidation, we compare experimental results with those of the multi-layer kinetics model described above. In all, the model has four adjustable parameters: the uptake coefficient, $\gamma_{\text{OH},i}$; the diffusion coefficient, $D_{\text{b},i}$; and the molar absorptivities of products B and C, $\varepsilon_{\text{B}}$, and $\varepsilon_{\text{C}}$, respectively.

At 60% RH, the experimental features can be reproduced (see Fig. 8) by setting $\gamma_{\text{OH},i}$ to 4.2, suggesting that oxidation

is very efficient. An uptake coefficient greater than unity indicates that each collision between OH and A leads to the formation of more than one molecule of B. In previous heterogeneous OH oxidation experiments, a wide range of OH uptake coefficients have been observed, including values greater than unity (George and Abbatt, 2010). In the particle phase, oxidation initiated by OH may form products directly from reactions with OH or indirectly from organic radicals, RO. The organic radicals may result from free radical chain reactions that have been shown to lead to very high effective uptake coefficients, in particular,

when species like NO and SO$_2$ are present to enhance conversion of RO$_2$ to RO (Richards-Henderson et al., 2015, 2016). We note that the value of $\gamma_{\text{OH},i}$ is dependent on the assumed average molecular mass, which is set to 326 g mol$^{-1}$, representative of a terphenyl oligomerization product of resorcinol, as discussed above. A greater molecular mass would result in a lower value of $\gamma_{\text{OH},i}$, so we cannot be sure whether free radical chain reactions are playing a role in the experiments. Furthermore, we assume that OH undergoes solely gas-surface reactions, following Shiraiwa et al. (2009). Recent measurements indicate that

this may be an over-simplification, as the reactive-diffusive length of OH in squalene is similar to that of ozone (Lee and Wilson, 2016), so surface-layer reactions may also occur. Nonetheless, we can state with confidence that the oxidation is very efficient.

We find that the observed trend in relative absorption at 60% RH cannot be reproduced if the aging particles are taken as well-mixed; if $D_{\text{b},i}$ is taken as 1 x 10$^{-14}$ cm$^2$ s$^{-1}$, the decay in the concentration of A is about the same in all layers (see Fig.

S6a), and the abrupt cessation of absorption enhancement cannot be captured. As shown in Fig. S7, if the absorptivities are fixed, and the uptake coefficient is scanned from 0.2 to 10, the trends in relative absorption have the same shape (simply stretched or compressed along the time axis), because the particles are well-mixed. On the other hand, if $D_{\text{b},i}$ is taken as 1 x 10$^{-16}$ cm$^2$ s$^{-1}$, the fraction of species A at the surface decays significantly faster than in the bulk (see Fig. S6c), the modelled



trends cannot be superimposed by scaling along the time axis (see Fig. S8), and the abrupt cessation of absorption enhancement can be captured. At 60% RH, secondary organic material derived from the gas-phase photo-oxidation of toluene has a diffusion coefficient on the order of $10^{-10}$ cm$^2$ s$^{-1}$ (Song et al., 2016). Though the present diffusion coefficient is surprisingly low, it is possible that the secondary organic material formed from resorcinol in the aqueous phase is primarily oligomeric and

comparatively viscous even at 60% RH. The same diffusion coefficient is assumed for all three species, but in reality $D_{b,i}$ may increase in going from A to B and decrease in going from B to C; if C is a product of fragmentation, it might be expected to have a greater diffusion coefficient in A than the self-diffusion coefficient of A. Indeed, plasticization has been shown to occur during heterogeneous OH oxidation of semi-solid alkane particles at high gas-phase OH concentrations (Wiegel et al., 2017).

Finally, the experimental features are reproduced by setting the molar absorptivities of B and C ($\varepsilon_B$ and $\varepsilon_C$) to 2.5$\varepsilon_A$

and 0, respectively. We note that this combination of parameters is likely one of several possible solutions. Even for a more tightly-constrained set of experimental data regarding consumption of oleic acid by heterogeneous reactions with ozone (Ziemann, 2005), more than one model scenario has been found to adequately reproduce observations of the number of oleic acid molecules in the particle phase (Pfrang et al., 2010).

At 15% RH, the experimental features are reproduced (see Fig. 8) by setting $\gamma_{OH,i}$ and $D_{b,i}$ to 1.0 and 1 x $10^{-18}$ cm$^2$

s$^{-1}$, respectively. The decrease in $\gamma_{OH,i}$ from 4.2 to 1.0 may be the result of differences in the distribution of initial BrC species in the particle or the orientation of the species at the surface. The decrease in $D_{b,i}$ by two orders of magnitude is reasonable, considering the significant RH-dependence of viscosity observed for secondary organic material derived from toluene and isoprene (Song et al., 2015, 2016). An upper limit to the diffusion coefficient of secondary organic material derived from toluene about $10^{-17}$ cm$^2$ s$^{-1}$ was found for RH ≤ 17% (Song et al., 2016). Based on the absence of bleaching during the

experimental timescale, we set $\varepsilon_c$ to be the same as $\varepsilon_B$; i.e., 2.5$\varepsilon_A$ (see Fig. S9). This equivalence may indicate that, if fragmentation occurs due to reaction of OH with B, the strict confinement of the fragments leads to recombination, such that the absorption persists. It is also possible that stable fragmentation products form, but they volatilize out of the condensed phase to a much greater extent than they diffuse to the bulk phase, because the particles are so viscous; in this case, the decreased molar absorptivity of the products would have little effect on the total particle absorption. In fact, a slight decrease

in the mean geometric surface diameter suggests that there is some degree of volatilization. On a related note, because the viscosity is so high, the products B and C are much more concentrated at the surface than in the bulk layers (see Fig. 9). Consequently, at 15% RH, the aged particles likely consist of unchanged BrC cores encased in thin, more-absorptive shells.

In general, we emphasize that the set of molecular properties presented above, which best fit the experimental data, were selected after multiple trials with different input parameter values and different model scenarios, as well. Although the

mechanism and final parameter set fit the data remarkably well, we acknowledge that the complex nature of the inherent chemistry, with changes in optical properties occurring alongside concentration, suggest that this parameter set should be viewed only semi-quantitatively; i.e., the solution is most useful for substantiating the mechanism leading to the changes in absorption.



## 4 Conclusions and atmospheric implications

In this study, we have demonstrated for the first time that secondary BrC aerosol derived from a phenolic precursor is susceptible to further photo-chemical aging after cloud processing and droplet evaporation. Specifically, at 60% RH, OH exposure induced rapid absorption enhancement followed by relatively slow bleaching of the surrogate BrC aerosol; at 15%

RH, OH exposure induced only slow absorption enhancement. Moreover, we have constructed a multi-layer kinetics model that captures the general features of the evolution of the optical properties of the particles. The candidate parameters suggest that the oxidation is very efficient, possibly even involving free radical chain reactions, and the surrogate BrC aerosol is very viscous. Free radical chain reactions may be more important in the atmosphere, where a lower concentration of OH results in a lower concentration of $RO_2$ and a reduced probability of chain termination reactions. Furthermore, the presence of other

pollutants like NO may enhance the conversion of $RO_2$ to RO (Richards-Henderson et al., 2015). Since the BrC is very viscous, it is important that the experimental timescale approximates the atmospheric timescale, so species are allowed sufficient time to diffuse within the particles. In the laboratory, flow-tube experiments with higher OH concentrations and shorter timescales than photo-oxidation chamber experiments may not as accurately account for free radical chain reactions and diffusion timescales.

Using our measurements and those of others, we can now speculate on the photochemical behaviour of BrC in the atmosphere. Recently, Sumlin et al. (2017) observed bleaching due to heterogeneous OH oxidation of primary BrC derived from biomass burning, which lost almost 50% of its absorption at 375 and 405 nm after the equivalent of about 4.5 days in the atmosphere. In our experiments at 60% RH, the SSA of the BrC first increased and then decreased to its initial value after about 2 h of photo-oxidation in the chamber; in the atmosphere, this would correspond to about 30 h, because the average OH

concentration in the chamber (1.6 x $10^7$ molecule $cm^{-3}$) is considerably greater than typical OH concentrations in the atmosphere (on the order of $10^6$ molecule $cm^{-3}$). In other words, the initial, rapid absorption enhancement could compensate for more than a day of bleaching. In the atmosphere, then, we may expect highly variable evolution of biomass burning plumes. If little or no secondary BrC forms, bleaching of primary BrC will dominate, and the absorption will decay uniformly. Indeed, Forrister et al. (2015) have observed such a trend in the field. If considerable secondary BrC forms, e.g., by cloud processing,

some additional absorption will develop. We have shown that additional absorption may develop after cloud processing, as well. These processes will compete with the bleaching of primary BrC, such that the total BrC absorption may persist for longer periods. Indeed, Zhang et al. (2017) observed that the absorption by BrC at 365 nm was largely preserved during convection and one day of residence in the upper troposphere. Our results suggest that the persistence of BrC in this field study was the result of absorption enhancement compensating for bleaching, rather than recalcitrance of the BrC. We speculate that,

upon further aging of this plume, bleaching would begin to play a greater role than absorption enhancement. In the 15% RH experiment, only absorption enhancement was observed after the equivalent of about 40 h in the atmosphere. Though such low RH conditions are less widely applicable in the atmosphere, there may be scenarios in which bleaching due to heterogeneous OH oxidation does not occur even days after secondary BrC formation from cloud processing of phenolic species.




The aerosol studied here is a reasonable proxy for the secondary BrC that may form in the atmosphere upon evaporation of cloud droplets. Biomass burning emits BC, primary BrC, and non-absorbing organic compounds, in both the gas and particle phases. These organic compounds include phenolic species, derived from the decomposition of lignin during combustion (Simoneit, 2002). Resorcinol is a representative phenolic emission of biomass burning (Simoneit, 2002; Veres et al., 2010; Wang et al., 2009); for example, Schauer et al. (2001) observed about 50 $\mu$g of resorcinol from the combustion of 1 kg of wood. Furthermore, about 95% of the resorcinol was in the particle phase, compared to about 55 and 0% for the 1,4- and 1,2-dihydroxybenzene isomers, respectively. Whether in the aerosol phase or in cloud droplets, resorcinol and other phenolic species may react with OH to form chromophores, like the bi- and terphenyl products discussed above. Interestingly, biphenyls have recently been identified in ambient cloud water (Cook et al., 2017). The formation of secondary BrC from OH oxidation of phenolic species during the day contrasts with some other routes of secondary BrC formation that occur at night and result in chromophores – e.g., pyruvic acid polymers (Rincón et al., 2009) and imine compounds (Zhao et al., 2015) – that may be bleached comparatively quickly. The photochemical behaviour of these and other classes of secondary BrC chromophores plays an important role in the overall climate effects of BrC, and similar experiments should be performed for other surrogates in the future.

*Acknowledgements*. This research was funded by the Natural Sciences and Engineering Research Council of Canada (NSERC). E. G. Schnitzler gratefully acknowledges a Postdoctoral Fellowship from NSERC.

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



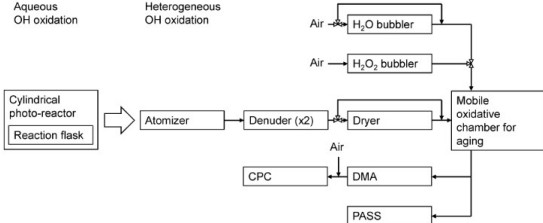

**Figure 1. Experimental setup during aqueous and heterogeneous OH oxidation. CPC: condensational particle counter; DMA: differential mobility analyzer; PASS: photo-acoustic spectrometer.**





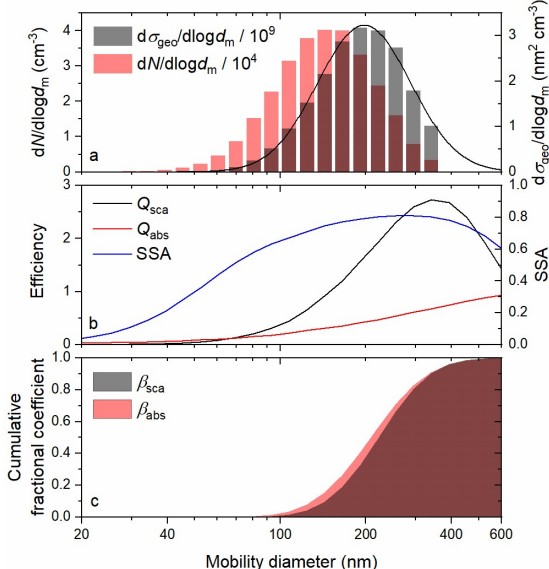

**Figure 2. Representative (a) number- and geometric-cross-section-weighted size distributions, (b) predicted scattering and absorption efficiencies and SSA, based on an assumed complex refractive index of $m = 1.35 + 0.41i$, and (c) cumulative fractional scattering and absorption coefficients, based on the observed size distribution and predicted efficiencies.**





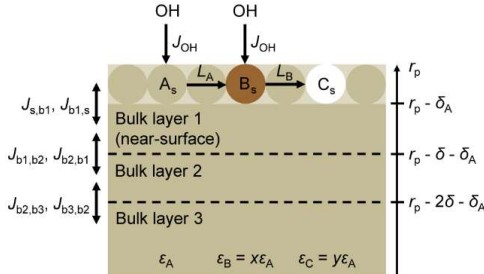

**Figure 3.** Schematic of the multi-layer kinetics model of heterogeneous OH oxidation.





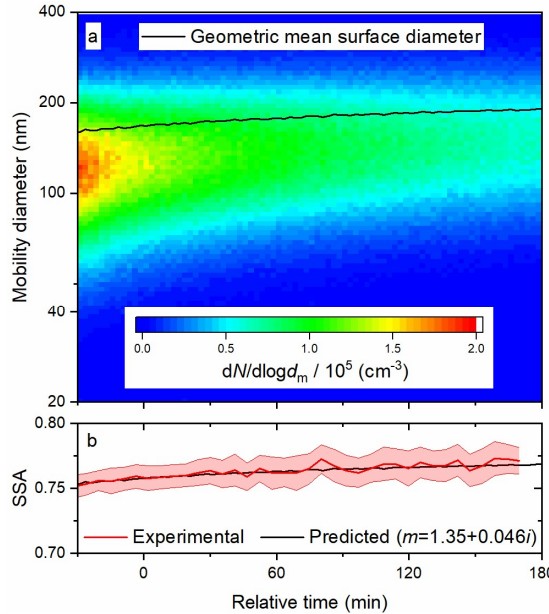

**Figure 4.** Time series of (a) size distribution and geometric mean surface diameter and (b) predicted (based on size distributions) and observed SSA during a deposition experiment at 60% RH. In (b), the upper and lower bounds illustrate one standard deviation about the 5-minute averages.





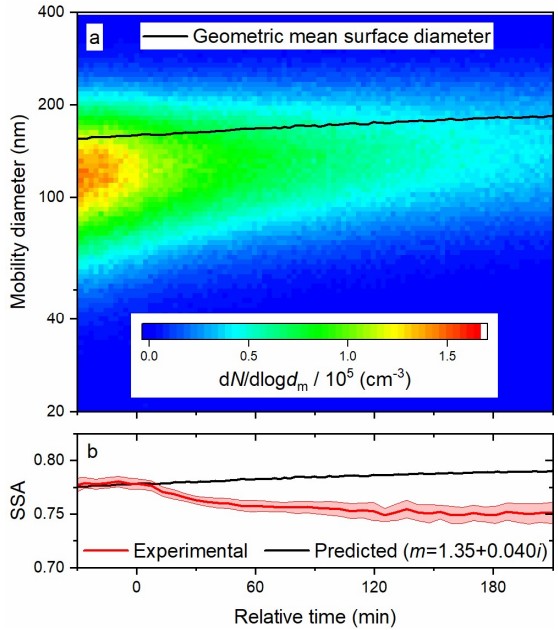

**Figure 5.** Time series of (a) size distribution and geometric mean surface diameter and (b) predicted (based on size distributions) and observed SSA during a photolysis experiment at 60% RH. In (b), the upper and lower bounds illustrate one standard deviation about the 5-minute averages.





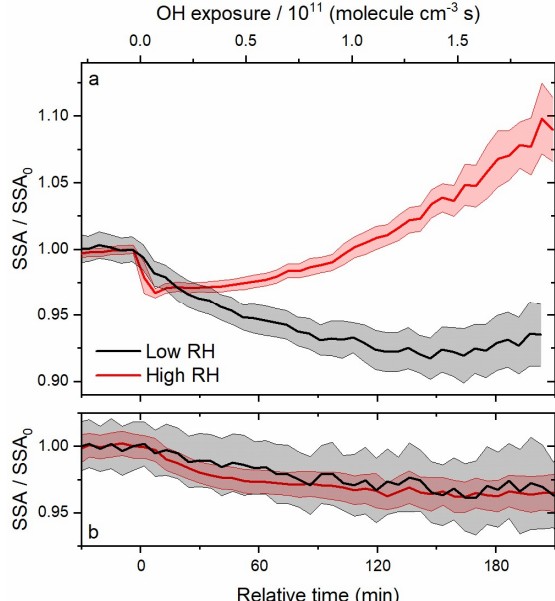

**Figure 6. Time series of relative SSA during (a) heterogeneous OH oxidation and (b) photolysis experiments at 15 and 60% RH.**





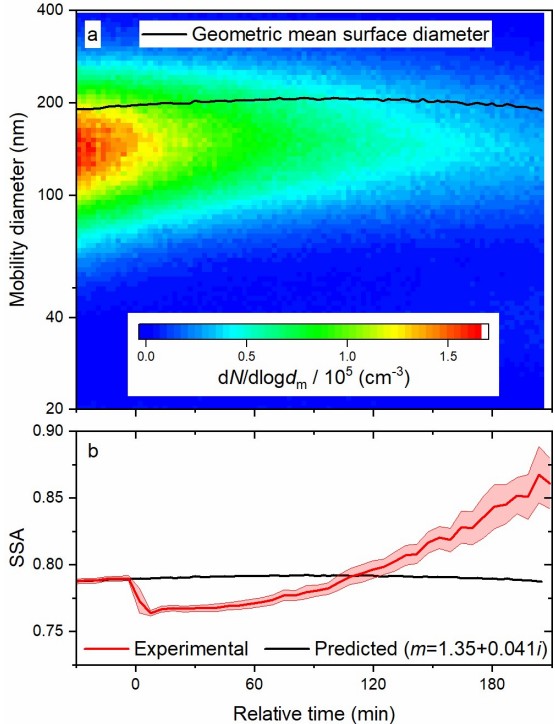

Figure 7. Time series of (a) size distribution and geometric mean surface diameter and (b) predicted (based on size distributions) and observed SSA during a photo-oxidation experiment at 60% RH. In (b), the upper and lower bounds illustrate one standard deviation about the 5-minute averages.





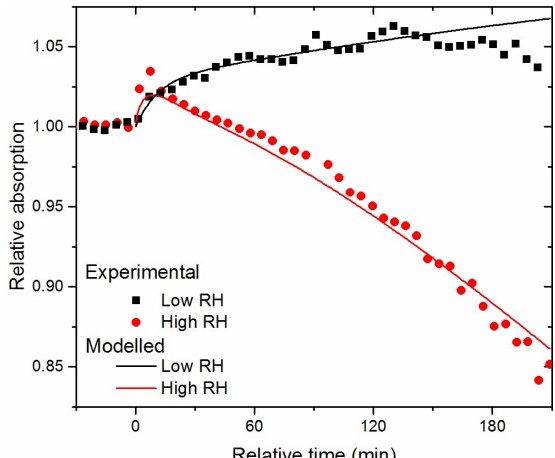

**Figure 8. Time series of observed and modelled relative absorption at 15 and 60% RH. The experimental trends are the inverse of relative SSA.**



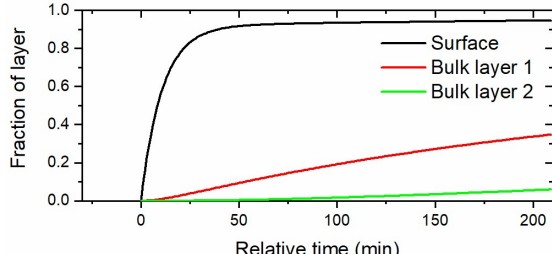

**Figure 9. Modelled fraction of surface and bulk layers 1 and 2 composed of products B and C at 15% RH.**