# Peer review of "Heterogeneous OH oxidation of secondary brown carbon aerosol"

_Atmospheric Chemistry and Physics, 2018_

## Referee Comment (RC1) · Anonymous Referee #1 · 30 Apr 2018

This manuscript reports experiments, in which the evolution of brown carbon (BrC) aerosol upon exposure to OH is followed by the optical properties (scattering and absorption) at low (15%) and higher (60%) relative humidity. BrC aerosol was produced from the aqueous photooxidation of solutions containing resorcinol and $H_2O_2$, thus resembling aged biomass burning aerosol with high aromaticity. The results are that at 60% RH, oxidation of this BrC aerosol first induced an enhancement of absorption, followed by bleaching, with an inverse behavior observed for the single scattering albedo (SSA). At 15% RH, only a slowly increasing absorption was observed during the time scale of the experiments. Interpretation of the results is facilitated by a multilayer kinetics model, in which chemistry is lumped into a simple oxidation scheme involving one parent BrC leading to one second and one third generation oxidation product with

differing optical properties. Comparison to experimental data in terms of optical properties, indicate that strongly contrasting diffusivity must be assumed between 15% RH and 60% RH to reproduce the experimental data. This allows speculating about different pathways of oligomerization and fragmentation to occur at various time periods. The evolution of BrC properties is a highly relevant topic of atmospheric aerosol chemistry due to ubiquitous presence of BrC compounds in a large variety of primary, aged primary or secondary organic aerosol.

The experiments seem to be well performed and carefully analyzed. Proper control experiments are performed to distinguish between photolysis and OH oxidation. Since the experiments are not accompanied by more detailed chemical analysis, the application of the kinetic model remains poorly constrained, though it provides a useful link between expected chemical processes and the optical properties, as they evolve under different humidity and thus likely differing diffusivity.

The manuscript is well written and structured; the conclusions are adequately supported by the experimental findings; and the kinetic model is presented and used with care and proper caveats. I recommend publication of this work with maybe just a few small revisions, following some specific comments below.

1) The model is based on chemical reactions just occurring at the surface, and the bulk only serves as a medium for reactants and products to diffuse; this seems reasonable for the reaction with OH. However, second generation oxidation may involve O2 or other reactive oxygen species deriving from the first and second step and may also proceed in the bulk. Of course, considering such would rapidly lead to more variables that would need to be tuned and would make the results more ambiguous. But maybe the authors could make an attempt in checking the sensitivity of the model results and parameters towards the experimental observables. I would also expect that O2 has quite different diffusivity than the large aromatic oligomers.

2) The estimated diffusivity at 60% RH and also the fact that diffusion limitations are

so apparent are a bit surprising. The authors are explaining it with the high aromatic content and the ease with which aromatic oligomers are formed. Can the hygroscopic growth be estimated from the experiment between 15% and 60% RH. The lack of significant water uptake could support the semi-solid character of these particles at 60% RH.

3) Based on the reported results, under the conditions of the experiments, OH oxidation dominated the changes in optical properties in comparison to pure photolysis alone. Could the authors try estimating the relative impact of photolysis and OH under atmospheric conditions. Photolysis of BrC or reactions of their triplet excited states may also lead to later generation radical processes, similar to those initiated by OH; therefore the relative impact of OH versus that of BrC induced photochemistry on aerosol aging may require some attention.

---

## Referee Comment (RC2) · Anonymous Referee #2 · 30 May 2018

Overall, I find this to be an interesting study that looks at how photochemical aging influences the absorptivity of aqueous, secondary brown carbon. The results and interpretation, if correct, are a useful contribution to the literature. I do, however, have two substantial concerns. (i) The photoacoustic method has been experimentally shown to have potential negative biases at elevated RH, despite the references given. (The authors missed a critical reference.) (ii) It is, at times, difficult to understand exactly what conditions were run for the optimized modeling, and thus it was a little difficult to fully understand the interpretation provided.. A table and further description may be helpful.

Abstract: It would be good to explicitly state that this study investigates heterogeneous processing of "aqueous, secondary BrC" or something like that, to distinguish from primary BrC.

[Figure]

I am concerned that there is a potentially fatal flaw in this study at least with respect to a portion of the data. It may be that there is not, but this needs to be addressed. The authors made some of their absorption measurements with their PAS instrument at elevated RH. They cite two studies saying that they "assume there is no evaporation of water..." and given two citations, both theoretical. Unfortunately, much more recent experimental evidence has developed that suggests that there can be negative biases that result from evaporation of water vapor. The key paper is by Langridge et al. (AS&T, 2013). The authors do not cite this paper, which is an unfortunate oversight. Unless the authors can demonstrate that their measurements at elevated RH are not impacted by evaporation of water vapor. The apparent bleaching that is observed here could, potentially, simply be a reflection of the particles becoming more hygroscopic upon oxidation, and thus there being a negative bias of increasing magnitude. I believe it is up to the authors to demonstrate that their results are not biased by evaporation effects. If they cannot, then the 60% RH observations should probably be removed.

Abstract: Regarding the conclusion that at 15% RH the particles are viscous enough to "confine products of fragmentation," if the products are confined, how does the OH reach these molecules in the first place to react with them? The high viscosity would similarly cause the reactions to occur primarily at the surface, correct? And if so, the products would be in a very good spot for evaporation.

P8/L11: Was only the SSA matched, or were the absolute absorption and scattering also matched during the RI determination? If only the SSA, how can the authors ensure that they have a unique solution? There are a multitude of combinations of n and k that can give the same SSA value. Especially given that the n value determined differs so much from other SOA types.

P9/L24: Presumably, a difference between 0.040 and 0.041 are within experimental uncertainty.

P10/L3: it would be helpful if the authors could clarify what they mean when they say

they "observed uniform bleaching." Also, the origin of the red curve in Fig. S5 is a bit unclear, if it does assume a constant n and k.

P10/L21: How large of a MW would be necessary to yield a gamma = 1?

Fig. 8: This figure could probably benefit from additional panels showing the time-evolution of the normalized concentrations of A, B and C. It is difficult to visualize in the model how there is a sudden turnover that occurs with the 60% RH experiment with only two products and a continuous evolution. It would seem, at least to me, that some step change is necessary and the origin of this is not abundantly clear from the text.

Figures S7/8/9/etc.: It would be very useful if the authors would report the epsilon values assumed for each of these simulations in the caption. It is difficult to understand whether the differences between e.g. Fig. S8 and S9 are the result of differences in the diffisivity alone, or because different assumptions have been made about the evolution of the absorptivity and the absorptivity of the products. I say this because the authors could do a better job of explaining how the evolution of the product species differs between the 60% RH and 15% RH cases, given that they show that at 60% RH the particles are already fairly viscous, with differential oxidation between the surface layers and the bulk. What happens if all the authors do is drop the diffusivity from $10^{-16}$ (which worked for RH = 60%) to $10^{-18}$, keeping everything else the same? Perhaps this is what Fig. S8 and Fig. S9a are showing, but it is not abundantly clear as written and presented. And I actually think that these figures fundamentally differ in what is assumed about the absorptivity of the products.

P12/L16: it might be good to indicate that Sumlin investigated only a handful of biomass sources.
* * *

---

## Author Comment (AC1) · 11 Jul 2018

We thank the reviewers for their helpful comments. In the process of addressing these comments, we have performed supporting experiments and improved the presentation and clarity of the manuscript. Our detailed responses to the individual comments are shown below; quotations from the manuscript are shown with changes in bold. We give line numbers from the tracked-changes manuscript where appropriate.

**Reviewer 1**

1.1    This manuscript reports experiments, in which the evolution of brown carbon (BrC) aerosol upon exposure to OH is followed by the optical properties (scattering and absorption) at low (15%) and higher (60%) relative humidity. BrC aerosol was produced from the aqueous photooxidation of solutions containing resorcinol and H2O2, thus resembling aged biomass burning aerosol with high aromaticity. The results are that at 60% RH, oxidation of this BrC aerosol first induced an enhancement of absorption, followed by bleaching, with an inverse behavior observed for the single scattering albedo (SSA). At 15% RH, only a slowly increasing absorption was observed during the timescale of the experiments. Interpretation of the results is facilitated by a multilayer kinetics model, in which chemistry is lumped into a simple oxidation scheme involving one parent BrC leading to one second and one third generation oxidation product with differing optical properties. Comparison to experimental data in terms of optical properties, indicate that strongly contrasting diffusivity must be assumed between 15% RH and 60% RH to reproduce the experimental data. This allows speculating about different pathways of oligomerization and fragmentation to occur at various time periods. The evolution of BrC properties is a highly relevant topic of atmospheric aerosol chemistry due to ubiquitous presence of BrC compounds in a large variety of primary, aged primary or secondary organic aerosol.

The experiments seem to be well performed and carefully analyzed. Proper control experiments are performed to distinguish between photolysis and OH oxidation. Since the experiments are not accompanied by more detailed chemical analysis, the application of the kinetic model remains poorly constrained, though it provides a useful link between expected chemical processes and the optical properties, as they evolve under different humidity and thus likely differing diffusivity.

The manuscript is well written and structured; the conclusions are adequately supported by the experimental findings; and the kinetic model is presented and used with care and proper caveats. I recommend publication of this work with maybe just a few small revisions, following some specific comments below.

Thank you for these positive comments.

1.2    The model is based on chemical reactions just occurring at the surface, and the bulk only serves as a medium for reactants and products to diffuse; this seems reasonable for the reaction with OH. However, second generation oxidation may involve O2 or other reactive oxygen species deriving from the first and second step and may also proceed in the bulk. Of course, considering such would rapidly lead to more variables that would need to be tuned and would make the results more ambiguous. But maybe the authors could make an attempt in checking the sensitivity of the model results and parameters towards the experimental observables. I would also expect that O2 has quite different diffusivity than the large aromatic oligomers.

We agree that radical products of the initial OH reaction likely form RO$_2$ species by reaction with O$_2$. These RO$_2$ species may be converted to RO and initiate further oxidation of the brown carbon constituents or react with each other in a termination step. They may also facilitate auto-oxidation by causing hydrogen shifts. As the reviewer points out, an attempt to model these processes would introduce other parameters, such as the diffusion coefficient of O$_2$ in the particles and the absorptivities of any additional products, so we have not added them into our simple multi-phase model. However, we have modified the manuscript to discuss the potential role of RO$_2$ species in more detail. For example, RO$_2$ initiated reactions are suggested to contribute to uptake coefficients greater than unity at 60% RH. We have modified the text as follows:

- **"Processes other than directly OH-initiated oxidation also occur. For example, radical products of the initial OH reaction likely form RO2 species by reaction with O2. These RO2 species may react with each other in a termination step or react with NO, for example, to form RO (Richards-Henderson et al., 2015, 2016), which may initiate further oxidation of the brown carbon constituents. They may also facilitate auto-oxidation by abstracting hydrogen from adjacent groups on the same molecule. We do not consider these processes in our simple multi-phase kinetics model, since many other parameters, such as the diffusion coefficient of O2 in the particles and the absorptivities of any additional products, would have to be introduced, but they may play a role in the apparent values of uptake coefficients."** (page 7, lines 20-26)

We take this opportunity to discuss a change to Figure 8. In the original manuscript, the modeled relative absorbance was compared to the inverse of the experimental relative single scattering albedo (SSA). Since the scattering coefficient appears in both the numerator and denominator of SSA and is much larger than the absorption coefficient, this comparison is problematic. To show the changing absorption properties of the particles more clearly, we now compare the modeled relative absorbance to the experimental absorption coefficient divided by that which would be expected if the initial complex refractive index ($m$) did not change during the experiment, accounting for size dependence. The experimental trends, shown in Figure R1, are also corrected for the contribution of photolysis.

[Figure]

Figure R1. Experimental and modelled trends in relative absorption.

At 60% RH, the final value of the plotted quantity is about 0.40, rather than the value of 0.85 using the comparison plotted in the original Figure 8. To reproduce this trend – in particular, the more significant bleaching – the diffusion coefficient must be increased

from 1 x $10^{-16}$ to 1 x $10^{-14}$ cm$^2$ s$^{-1}$, at which the particle is well-mixed, not semi-solid. The uptake coefficients and absorptivity were changed slightly, but do not alter the original discussion. At 15% RH, the diffusion coefficient was also increased by two orders of magnitude. At both RH conditions, the product C is set to be non-absorbing, but at 15% RH the uptake coefficient associated with its formation is set to only 0.1. The text was modified by substituting the new parameter values throughout. Other changes include the following:

- **"The parameters also suggest that, as RH decreases, reactivity decreases, and aerosol viscosity increases, such that particles are well-mixed at 60% RH but not at 15% RH**." (page 1, lines 19-22)

- **"This modelled relative absorption is compared to the to the experimental absorption coefficient normalized to that which would be expected if the initial complex refractive index (*m*) did not change during the experiment, accounting for size dependence. Experimental times series of this normalized absorption were derived for photolysis and heterogeneous OH oxidation experiments, and the latter were corrected for the effect of photolysis.** ~~During heterogeneous OH oxidation, the evolution of the SSA of the particles incorporates changes in both absorption and scattering. However, the changes in absorption are likely dominant. For example, for fixed values of n (1.35) and particle diameter (200 nm), a 50% decrease in k (from 0.04 to 0.02) gives a 10% increase in Qsca but almost a 50% decrease in Qabs. Consequently, we assume that Qsca is steady, and we compare the modelled trends in relative absorption to the experimental trends in the inverse of relative SSA.~~" (page 7, lines 11-19)

- We find that the observed trend in relative absorption at 60% RH **– in particular, the significant bleaching –** can be reproduced **only** if the aging particles are taken as well-mixed**, so diffusion of A from the bulk to the surface is not restricted**~~; if $D_{b,t}$ is taken as 1 x $10^{-14}$ cm$^2$ s$^{-1}$, the decay in the concentration of A is about the same in all three bulk layers (see Fig. S6a), and the abrupt cessation of absorption enhancement cannot be captured. As shown in Fig. S7, if the absorptivities are fixed, and the uptake coefficient is scanned from 0.2 to 10, the trends in relative absorption have the same shape (simply stretched or compressed along the time axis), because the particles are well-mixed. On the other hand, if $D_{b,t}$ is taken as 1 x $10^{-16}$ cm$^2$ s$^{-1}$, the fraction of species A at the surface decays faster than in the bulk (see Fig. S6c), the modelled trends cannot be superimposed by scaling along the time axis (see Fig. S8), and the abrupt cessation of absorption enhancement to be captured~~." (page 12, lines 8-15)

- **"Though the extent of bleaching is captured by the model, and rapid initial colour enhancement is not. We speculate the other processes, like auto-oxidation, could contribute to this feature."** (page 12, lines 20-22)

1.3    The estimated diffusivity at 60% RH and also the fact that diffusion limitations are so apparent are a bit surprising. The authors are explaining it with the high aromatic content and the ease with which aromatic oligomers are formed. Can the hygroscopic growth be estimated from the experiment between 15% and 60% RH. The lack of significant water uptake could support the semi-solid character of these particles at 60% RH.

As described above, the modelled diffusion coefficient at 60% RH increased by two orders of magnitude, upon comparing the modelled relative absorbance to the experimental absorption coefficient divided by that which would be expected of a constant refractive index. Consequently, the particles are no longer estimated to be semi-solid at 60% RH.

Nonetheless, we agree that it would be informative to roughly estimate the extent of hygroscopic growth at 60% RH. Unfortunately, the initial size distributions vary between experiments, including replicates at the same RH, so we cannot directly compare the 15 and 60% RH experiments to evaluate hygroscopic growth at 60% RH.

To address this issue, we performed an additional experiment, in which BrC particles were sampled from the chamber alternately with and without a diffusion dryer upstream of the DMA. The conditioning was alternated every 15 min, allowing time for the RH of the sheath flow to stabilize. As shown in Figure R2 (also Figure S5 in the supporting information), the size distribution slowly shifts to larger mobility diameters, because smaller particles are lost faster than larger particles. Besides this trend, the variation between the conditions appears to be negligible, so there is not significant water uptake. Nonetheless, slight differences in scattering coefficients of nascent and dried particles are observed, as described in our reply to comment 2.3.

[Figure]

Figure R2. Size distributions of dried and nascent particles, collected alternately in 15 min intervals.

The manuscript was modified as follows:

- **"In one experiment, BrC particles were sampled from the chamber alternately with and without a diffusion dryer upstream of the DMA. The conditioning was alternated every 15 min, allowing time for the RH of the sheath flow to stabilize. As shown in Fig. S6, the size distribution slowly shifts to larger mobility diameters, because smaller particles are lost faster than larger particles. Besides this trend, the variation between the conditions appears to be negligible, so there is not significant water uptake at 60% RH."** (page 9, lines 6-10)

1.4     Based on the reported results, under the conditions of the experiments, OH oxidation dominated the changes in optical properties in comparison to pure photolysis alone. Could the authors try estimating the relative impact of photolysis and OH under atmospheric conditions. Photolysis of BrC or reactions of their triplet excited states may also lead to later generation radical processes, similar to those initiated by OH; therefore the relative impact of OH versus that of BrC induced photochemistry on aerosol aging may require some attention.

This is a very important issue, and we now address it in the text as follows:

- "The emission of the UV-B black-lights in the chamber, **with a peak at 310 nm,** is not representative of natural sunlight, so we do not draw direct comparisons between the timescale of our experiments and that of photolysis in the atmosphere. Rather, we performed photolysis experiments to account for the effects of photolysis during the heterogeneous OH oxidation experiments. **Wong et al. (2015) have shown that for this chamber equipped with UV-B bulbs, the photo flux at wavelengths below 310 nm is much greater than for a clear-sky summer day; for example, at 300 nm, the photon flux from the bulbs is close to its peak value of about $1 \times 10^{14}$ photons s$^{-1}$ cm$^{-2}$ nm$^{-1}$, while the photon flux outside is negligible. Light at wavelengths below 310 nm is likely driving most of the photo-chemistry during the photolysis experiments, so we believe the relative impact of photolysis would be small in the atmosphere, even considering that the ambient OH concentration is lower than in the chamber."** (page 9, lines 13-21)

---

## Author Comment (AC2) · 11 Jul 2018

We thank the reviewers for their helpful comments. In the process of addressing these comments, we have performed supporting experiments and improved the presentation and clarity of the manuscript. Our detailed responses to the individual comments are shown below; quotations from the manuscript are shown with changes in bold. We give line numbers from the tracked-changes manuscript where appropriate.

**Reviewer 2**

2.1     Overall, I find this to be an interesting study that looks at how photochemical aging influences the absorptivity of aqueous, secondary brown carbon. The results and interpretation, if correct, are a useful contribution to the literature. I do, however, have two substantial concerns. (i) The photoacoustic method has been experimentally shown to have potential negative biases at elevated RH, despite the references given. (The authors missed a critical reference.) (ii) It is, at times, difficult to understand exactly what conditions were run for the optimized modeling, and thus it was a little difficult to fully understand the interpretation provided. A table and further description may be helpful.

> We agree that these are substantial concerns. To address the possibility of negative bias at 60% RH, we have performed a supporting experiment, described in detail in our reply to comment 2.3. We have added the critical reference to the manuscript. We have also modified and clarified our discussion of the modeling results, as described in our replies to comments 1.2 and 2.8-2.10.

2.2     Abstract: It would be good to explicitly state that this study investigates heterogeneous processing of "aqueous, secondary BrC" or something like that, to distinguish from primary BrC.

> Done.

2.3     I am concerned that there is a potentially fatal flaw in this study at least with respect to a portion of the data. It may be that there is not, but this needs to be addressed. The authors made some of their absorption measurements with their PAS instrument at elevated RH. They cite two studies saying that they "assume there is no evaporation of water ..." and given two citations, both theoretical. Unfortunately, much more recent experimental evidence has developed that suggests that there can be negative biases that result from evaporation of water vapor. The key paper is by Langridge et al. (AS&T, 2013). The authors do not cite this paper, which is an unfortunate oversight. Unless the authors can demonstrate that their measurements at elevated RH are not impacted by evaporation of water vapor. The apparent bleaching that is observed here could, potentially, simply be a reflection of the particles becoming more hygroscopic upon oxidation, and thus there being a negative bias of increasing magnitude. I believe it is up to the authors to demonstrate that their results are not biased by evaporation effects. If they cannot, then the 60% RH observations should probably be removed.

> The reviewer suggests that the particles may become more hygroscopic during oxidation, leading to more water uptake and, upon irradiation in the photo-acoustic spectrometer, water evaporation from the particles. Through evaporation, some of the energy of the absorbed photons would be lost without contributing to the detected pressure wave. In

addition to this instrumental artifact, progressively more water uptake could lead to genuinely greater scattering coefficients, which would also result in apparent bleaching.

To address these concerns, we performed an additional photo-oxidation experiment. Particles were sampled from the chamber to the photo-acoustic spectrometer and scanning mobility particle spectrometer, alternately with and without a diffusion dryer downstream of the chamber. Results of this experiment are illustrated in Figures R2-R3 and the supporting information (Figures S7 and S8 in the supporting information). We note that this experiment was conducted in a different chamber than the original experiments, so some of the conditions (like average OH concentration) may differ. Unlike in the original experiments, some particle growth occurred, leading to a significant increase in the scattering coefficient.

[Figure]

Figure R3. Time series of the absorption coefficient at 405 nm for dried (low RH) and nascent (high RH) particles, measured by alternatively sending the particle flow through a diffusion dryer.

[Figure]

Figure R4. Time series of the scattering coefficient at 405 nm for dried (low RH) and nascent (high RH) particles.

Nonetheless, the trends in absorption and scattering coefficients demonstrate that the results of the original experiments are valid. First, as shown in Figure R3, the decrease in the absorption coefficient following the initial colour enhancement can be well represented by fitting a single logistic function to measurements for both nascent and dried particles. In other words, there is not a significant negative bias from water evaporation. Second, as shown in Figure R4, the relative difference between the scattering coefficients of the nascent and dried particles, based on fitting exponential functions to the measurements after the initial particle growth, is roughly constant across four reaction time intervals from about 50 to 120 min: 11, 10, 11, and 13%. This consistency indicates that the particles are not taking up significantly more water as they react with OH.

We have modified the text as follows:

- "**One potential source of bias in photo-acoustic measurements is evaporation of water from particles (Baker, 1976; Raspet et al., 2003; Langridge et al., 2013); through evaporation, some of the energy of the absorbed photons may be lost and not contribute to the detected pressure wave. On this basis, the higher RH was selected to be lower than the maximum operating RH of the PASS (70%). To verify that evaporation of water did not influence the measurements of absorption coefficients, we alternately sampled with and without a diffusion dryer downstream the chamber in one experiment.**" (page 5, lines 26-30)

- "**As described above, evaporation of water from particles can result in a negative bias in photo-acoustic measurements Langridge et al., 2013). If heterogeneous OH oxidation significantly increased the hygroscopicity of the particles, the water content of the aerosol would increase during the experiment. The resulting increase in the magnitude of the bias could contribute to apparent bleaching. In addition to this instrumental artifact, progressively more water uptake could lead to genuinely greater scattering coefficients, which would also result in apparent bleaching. We investigated these potential effects by alternately sampling with and without a diffusion dryer downstream of the chamber at 60% RH. As shown in Fig. S7, the absorption coefficient does not depend on the conditioning. The scattering coefficient is about 10% lower for the dried particles (see Fig. S8), but this difference is roughly steady during photo-oxidation. In other words, the particles do not become significantly more hygroscopic, and the changes in absorption and scattering coefficients are indeed due to the chemical evolution of the particles.**" (page 11, lines 1-10)

We thank the reviewer for bringing the paper by Langridge et al. to our attention; we now cite this paper, along with the earlier theoretical papers.

2.4     Abstract: Regarding the conclusion that at 15% RH the particles are viscous enough to "confine products of fragmentation," if the products are confined, how does the OH reach these molecules in the first place to react with them? The high viscosity would similarly cause the

reactions to occur primarily at the surface, correct? And if so, the products would be in a very good spot for evaporation.

Under the conditions originally proposed for 15% RH – specifically, the very small diffusion coefficient of $1 \times 10^{-18}$ cm$^2$ s$^{-1}$ – the model products B and C, which form at the surface, do not diffuse appreciably. Even after 180 min, their concentrations in bulk layer 3 are negligible. This observation was the basis of our conclusion that the products are strictly confined to the surface. We agree that the products of fragmentation (i.e., species C) likely evaporate from the particle surface to some extent. In fact, the size distributions at 15% RH shift slightly to smaller sizes at later OH exposures. Since we now assume a larger diffusion coefficient of $1 \times 10^{-16}$ cm$^2$ s$^{-1}$, as described in our reply to comment 1.2, we have removed the suggestion that recombination may occur, as follows:

- " It is  possible that  fragmentation products  volatilize out of the condensed phase to a  greater extent than they diffuse to the bulk phase, because the particles are so viscous. In fact, a slight decrease in the mean geometric surface diameter suggests that there is some degree of volatilization." (page 13, lines 6-11)

2.5    P8/L11: Was only the SSA matched, or were the absolute absorption and scattering also matched during the RI determination? If only the SSA, how can the authors ensure that they have a unique solution? There are a multitude of combinations of n and k that can give the same SSA value. Especially given that the n value determined differs so much from other SOA types.

SSA, rather than the absolute absorption and scattering coefficients, was used to derive an approximate complex refractive index ($m$). In the deposition experiment, the aerosol is unperturbed, so $m$ is assumed to be constant, while the aerosol distribution and SSA change slightly. We agree that any reasonable combination of $n$ and $k$ can give good agreement for an individual distribution, but it cannot reproduce the observed trend in SSA, as shown in Figure R5 and the supporting information (Figure S4).

[Figure]

Figure R5. Time series of observed and predicted (based on the size distributions) SSA during a deposition experiment.

We agree that it would be preferable to use the absolute absorption and scattering coefficients. However, we find that the coefficients predicted from the distributions are lower than those observed for all values of $m$. Unfortunately, the impactor installed in the DMA had an orifice of 0.0502 cm, but the value was set to 0.0708 cm in the firmware, so the flow rate on the DMA display was too high. As a result, the particles were diluted more than expected before reaching the CPC, leading to lower number and cross sectional area concentrations.

We stress that, here, even an approximate value of the $m$ is adequate to distinguish changes in the SSA due to changes in the size distribution from those in the chemical composition. For all assumed $m$ values, for example, the sudden colour enhancement and significant bleaching observed at 60% RH is not predicted, so it must be due to chemical evolution.

The manuscript was modified as follows:

- **"Stepping $n$ from 1.40 to 1.55 and adjusting $k$ accordingly demonstrates that the good agreement for this refractive index is unique, as shown in Fig. S9. We stress that even an approximate value of the refractive index is adequate to distinguish whether changes in the SSA are due to changes in the size distribution or chemical composition."** (page 8, line 33-page 9, line 1)

2.6     P9/L24: Presumably, a difference between 0.040 and 0.041 are within experimental uncertainty.

We agree that such a small difference in $k$ is within the experimental uncertainty; nonetheless, the respective values give the best agreement with the average values of SSA. We modify the text, acknowledging the values are similar:

- "The initial geometric mean surface diameter was slightly higher than in the photolysis experiment described above (about 196 nm compared to 160 nm), so although the initial aerosol was slightly more scattering (higher SSA), the value of k required to reproduce the SSA before irradiation is  **very similar** (0.041 compared to 0.040)." (page 10, lines 24-27)

2.7     P10/L3: it would be helpful if the authors could clarify what they mean when they say they "observed uniform bleaching." Also, the origin of the red curve in Fig. S5 is a bit unclear, if it does assume a constant n and k.

We apologize for a mistake in the legend of the original Figure S5b: the labels for the red and black curves were swapped. Assuming constant values of *n* and *k* gives the relatively steady predicted curve shown in black. The observed uniform bleaching is evident, beginning at relative time zero, coincident with the cusp of the red curve. The labels have been corrected.

2.8     P10/L21: How large of a MW would be necessary to yield a gamma = 1?

With the current set of parameters for 60% RH, decreasing the uptake coefficient for the initial BrC (i.e., species B) from 5.0 to 1.0 requires increasing the molecular weight from 326 to 1600 g mol$^{-1}$, in order to reproduce the experimental trend in relative absorption. We modify the text as follows:

- "**Scaling the uptake coefficients such that $\gamma_{OH,A}$ decreases from 5.0 to 1.0 requires increasing the molecular weight from 326 to 1600 g mol$^{-1}$. It is difficult to rationalize such large oligomers forming during the 4-h aqueous photo-oxidation, so secondary radicals are indeed likely contributing to oxidation.**" (page 12, lines 1-3)

2.9     Fig. 8: This figure could probably benefit from additional panels showing the time-evolution of the normalized concentrations of A, B and C. It is difficult to visualize in the model how there is a sudden turnover that occurs with the 60% RH experiment with only two products and a continuous evolution. It would seem, at least to me, that some step change is necessary and the origin of this is not abundantly clear from the text.

We have added two panels to Figure 8, shown below as Figure R6, one for each RH condition. In each, 12 time series are plotted, one for each species in each layer. As shown in the middle panel, the concentrations of the individual species are the same in all four layers at 60% RH; in other words, the particles are well-mixed. In contrast, as shown in the bottom panel, the concentrations of the individual species vary significantly across the layers at 15% RH.

[Figure]

Figure R6. Time series of (a) observed and modelled relative absorption at 60 and 15% RH and the modelled relative concentrations of A, B, and C at (b) 60 and (c) 15% RH. The curves are shaded according to layer, with the darkest curves corresponding to the surface layer.

The text was modified as follows:

- "**As shown in Fig. 8b, the concentrations of A, B, and C are the same in all four layers.**" (page 12, line 14)

- "On a related note, because the viscosity is so high, the products B and C are much more concentrated at the surface than in the bulk layers (see Fig.  **8c**). Consequently, at 15% RH, the  **aging** particles likely **consist of less absorbing cores and highly absorbing shells** ." (page 13, lines 11-14)

2.10    Figures S7/8/9/etc.: It would be very useful if the authors would report the epsilon values assumed for each of these simulations in the caption. It is difficult to understand whether the differences between e.g. Fig. S8 and S9 are the result of differences in the diffisivity alone, or because different assumptions have been made about the evolution of the absorptivity and the absorptivity of the products. I say this because the authors could do a better job of explaining how the evolution of the product species differs between the 60% RH and 15% RH cases, given

that they show that at 60% RH the particles are already fairly viscous, with differential oxidation between the surface layers and the bulk. What happens if all the authors do is drop the diffusivity from 10^-16 (which worked for RH = 60%) to 10^-18, keeping everything else the same? Perhaps this is what Fig. S8 and Fig. S9a are showing, but it is not abundantly clear as written and presented. And I actually think that these figures fundamentally differ in what is assumed about the absorptivity of the products.

> To generate these figures, $\varepsilon_B$ was set to 2.5, and $\varepsilon_B$ was set to either zero or 2.5, as well. These are the same conditions that were used to give the original model results in Figure 8, so the supporting figures do not fundamentally differ from what was described about the absorptivities of the products in the text.

> However, Figures S7-S9 were intended to illustrate how a low diffusion coefficient was required in the model even at 60% RH. Comparing the modeled relative absorption to the experimental absorption coefficient divided by that which would be expected of a constant refractive index resulted in different values for the model parameters, as described in our reply to comment 1.2. In particular, the diffusion coefficient increased by two orders of magnitude, required to reproduce the significant bleaching. Consequently, we have decided to omit the original Figures S7-S9 from the revised manuscript.

2.11    P12/L16: it might be good to indicate that Sumlin investigated only a handful of biomass sources.

> We have modified the text as follows:

- "Recently, Sumlin et al. (2017) observed bleaching due to heterogeneous OH oxidation of primary BrC derived from  burning **of a number of types of biomass fuels**, which lost almost 50% of its absorption at 375 and 405 nm after the equivalent of about 4.5 days in the atmosphere." (lines )